# Digital Evaluation of the Accuracy of Computer-Guided Dental Implant Placement: An In Vitro Study

**Seong-Min Kim [1,2], Keunbada Son [2,3], Duk-Yeon Kim [2] and Kyu-Bok Lee [1,2,*]**

[1] Department of Prosthodontics, School of Dentistry, Kyungpook National University, 2177 Dalgubeol-daero, Jung-gu, Daegu 41940, Korea

[2] Advanced Dental Device Development Institute, Kyungpook National University, 2177 Dalgubeol-daero, Jung-gu, Daegu 41940, Korea

[3] Department of Dental Science, Graduate School, Kyungpook National University, 2177 Dalgubeol-daero, Jung-gu, Daegu 41940, Korea

[*] Correspondence: kblee@knu.ac.kr; Tel.: +82-053-600-7674

**Abstract:** Compared to traditional implant surgical guides, computer-assisted implant surgical guides can be considered for positioning implants in the final prosthesis. These computer-assisted implant surgical guides can be easily fabricated with personal 3D printers after being designed with implant planning CAD software. Although the accuracy of computer-assisted implant surgical guides fabricated using personal 3D printers is an important factor in their clinical use, there is still a lack of research examining their accuracy. Therefore, this study evaluated the accuracy of computer-assisted implant surgical guides, which were designed using two implant planning CAD software programs (Deltanine and R2gate software) and fabricated with personal 3D printers using a non-radiographic method. Amongst the patients who visited Kyungpook National University Dental Hospital, one patient scheduled to undergo surgery of the left mandibular second premolar was randomly selected. Twenty partially edentulous resin study models were produced using a 3D printer. Using the Deltanine and R2gate implant planning CAD software, 10 implant surgical guides per software were designed and produced using a personal 3D printer. The implants (SIII SA (Ø 4.0, L = 10 mm), Osstem, Busan, Korea) were placed by one skilled investigator using the computer-assisted implant surgical guides. To confirm the position of the actual implant fixture, the study models with the implant fixtures were scanned with a connected scan body to extract the STL files, and then overlapped with the scanned file by connecting the scan body-implant fixture complex. As a result, the mean apical deviation of the Deltanine and R2gate software was $0.603 \pm 0.19$ mm and $0.609 \pm 0.18$ mm, while the mean angular deviation was $1.97 \pm 0.84°$ and $1.92 \pm 0.52°$, respectively. There was no significant difference between the two software programs ($p > 0.05$). Thus, the accuracy of the personal 3D printing implant surgical guides is in the average range allowed by the dental clinician.

**Keywords:** dental implant; computer-aided design; implant surgical guide; additive manufacturing

## 1. Introduction

Digital dentistry has evolved from cone-beam computed tomography (CBCT), intra-oral scanners, computer-aided design (CAD) software, and computer-aided manufacturing (CAM), which has had a profound impact on dental implantology [1–3]. In particular, additive manufacturing (AM) technology, known as 3D printing, has contributed to the successful implementation of computer-guided implant surgery [1,3]. Traditional implant surgical guides with modified radiographic templates require

complex laboratory procedures which can be inaccurate, making it somewhat difficult to accurately position implants in their planned locations [4–9]. On the other hand, computer-assisted implant surgical guides can consider important anatomical structures, save time, and aid implant placement by drawing the pre-planned final prosthesis design [10–12].

Accurate computer-assisted implant surgical guides are designed with implant planning CAD software to process the information obtained from CBCT, intra-oral scanners, and diagnostic casts [13]. Several studies have previously measured the accuracy of computer-assisted implant surgical guides designed using various implant planning CAD software [14]. According to a systemic review of the accuracy of the implant surgical guides, the mean apical deviation between the planned position and placed position was 1.4 mm, and the angle deviation was 3.5° [15]. However, the deviations may vary in different studies [16].

Son's study shows that the positioning accuracy of computer-guided implants can vary depending on experimental conditions, and that the accuracy of a cone beam computed tomogram (CBCT) or intraoral scanning of the patient, and the accuracy of 3D printing or milling, have been reported to have a significant effect on the positioning of implants [17]. If the positioning accuracy of computer-guided implants is inaccurate, it can lead to an unintended position, which may be the biggest cause of implant failure.

The positioning accuracy of the placed implants is measured by superimposing the pre- and post-operative CBCT images. However, CBCT can be inaccurate due to resolution and distortion, and errors may occur during the superposition of the CBCT images. In addition, the presence of metal artifacts in the oral cavity may lower the resolution of the CBCT images [18,19]. A non-radiographic method has been recently introduced to obtain the implant position and measure the placed implant position, by using an intraoral scanner and superimposing it on the previously obtained implant fixture scan data [20]. This allows for evaluation of the accuracy of the placed implant without post-operative CBCT imaging.

In the 1980s, Charles Hull developed an early 3D printing device called the stereolithography apparatus (SLA) that could create 3D models from digital data [21]. Since the important patents of AM technology have recently expired, the 3D-printer market is further developing [22]. As a result, smaller 3D printers are being introduced into dental clinics quicker and with lower costs [23,24]. These 3D printers are called in-office or personal 3D printers [25,26]. Computer-assisted implant surgical guides can be easily produced with personal 3D printers after being designed with implant planning CAD software [27].

Although the accuracy of computer-assisted implant surgical guides produced by personal 3D printers is an important factor in their clinical use, there is still a lack of research examining their accuracy. Therefore, this study evaluated the accuracy of computer-assisted implant surgical guides, which were designed using implant planning CAD software (Deltanine and R2gate software) and fabricated with personal 3D printers using a non-radiographic method. The null hypothesis of this study is that there is no difference in the accuracy of surgical guides fabricated by both types of software.

## 2. Materials and Methods

Amongst the patients who visited Kyungpook National University Dental Hospital, one patient who was scheduled to undergo surgery of the left mandibular second premolar with an identifiable inferior alveolar nerve was randomly selected. Before placing the implant, we took a preliminary impression and created a diagnostic cast using hard plaster. A 3D model scanner (Freedom HD, Degree of Freedom, Seoul, Korea) was used to scan the diagnostic model and save the patient's intraoral soft tissue surface information in a Surface Tesselation Language (STL) file. The patient's hard-tissue information was obtained using CBCT (Alphard-3030, ASAHI Rogentgen, Kyoto, Japan) and stored as a Digital Imaging and Communication in Medicine (DICOM) file. Twenty partially edentulous resin study models were produced using a 3D printer (ZENITH, Dentis, Daegu, Korea). To replicate the cancellous bone on the mandibular second premolar alveolar bone, the resin on the opposite side of

the study models was removed from the surface of the alveolar bone, and the removed portion was hardened by filling it with orthodontic resin (Dentsply Sirona, York, PA, USA) and sawdust. Twenty partially edentulous resin study models were scanned (CS3600, Carestream Dental LLC, Atlanta, GA, USA) and the results were saved as standard tessellation language (STL) files.

Pilot experiments were conducted 2 times to determine the sample size, and 8 samples per software were calculated using a power analysis software (G*Power v3.1.9.2, Heinrich Heine University, Düsseldorf, Germany) (actual power = 95.05%; power = 95%; $\alpha$ = 0.05). To increase power, the number of samples presented in this study was determined to be 10 per software.

Using the Deltanine (Daesung, Seoul, Republic of Korea) and R2gate (Megagen, Daegu, Republic of Korea) implant planning CAD software; the CBCT DICOM file taken before placement, and the intraoral scanning STL file were retrieved and overlapped (Figure 1A), the implant position was planned (Figure 1B), and a computer-assisted implant surgical guide was produced (Figure 1C). Ten implant surgical guides per software were designed and fabricated using a personal 3D printer (*n* = 10) (Figure 1D).

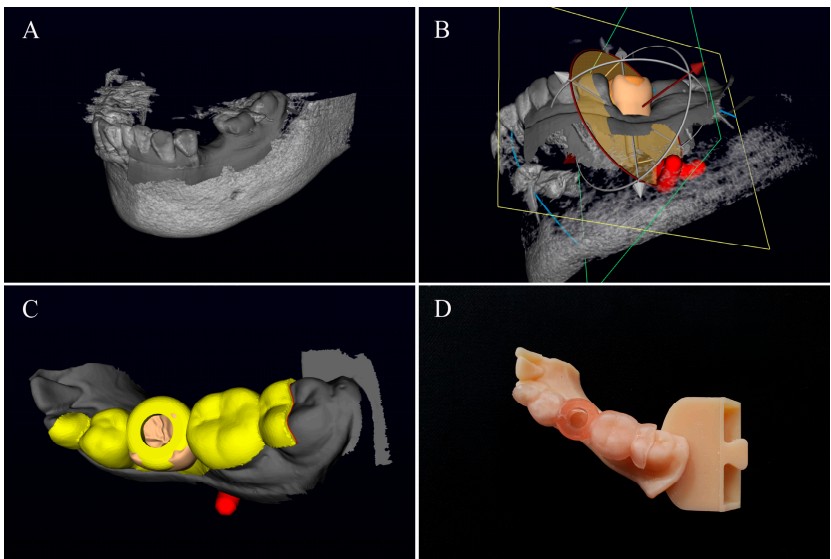

**Figure 1.** Process of surgical guide production. (**A**) Overlap between data. (**B**) Implant position plan. (**C**) Virtual surgical guide production. (**D**) Surgical guide production through a 3D printer.

The implants (TSIII SA (Ø 4.0, L = 10 mm), Osstem, Busan, Korea) were placed by one skilled investigator using the computer-assisted implant surgical guides and one guide surgical kit (Osstem, Busan, Korea). All implant fixture placement procedures were conducted by one skilled investigator.

The accuracy of the implant fixture placement angle was evaluated by measuring the angle of the straight line connecting the top and bottom of each implant fixture (Figure 2A). Accuracy of the implant fixture placement depth was confirmed by measuring the length of the straight line connecting the planned implant fixture apex and the placed implant fixture apex (Figure 2B).

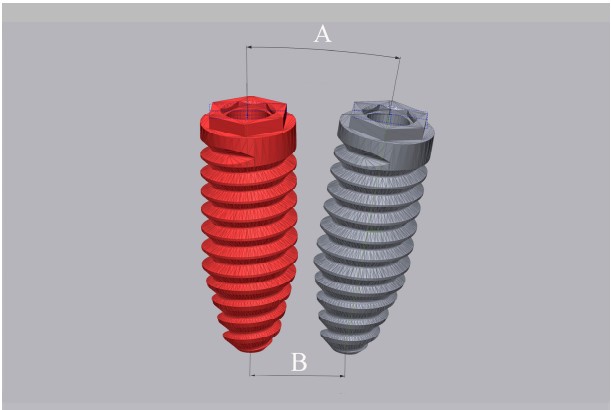

**Figure 2.** Measuring deviations between planned (red) and placed (gray) implants. (**A**) Angular deviation at the central axis of the implant. (**B**) Linear deviation at implant apex.

To evaluate the apical deviation and angular deviation, 3D inspection software (Geomagic control X, 3D system, Morrisville, NY, USA) was used, and the scan, design, and sleeve indicator STL files were extracted from the implant planning CAD software and overlapped to show the position of the planned implant fixture (Figure 3A). To confirm the position of the actual implant fixture, the study models with implant fixtures were scanned with a connected scan body (Osstem, Busan, Korea) to extract STL files, and were overlapped with the scanned file by connecting the scan body-implant fixture complex (Figure 3B). Precise scan data was obtained by calibration of the 3D scanner, and the operator performed the evaluation by confirming the exact overlapping in the 3D inspection software. A virtual cylinder was created for each implant fixture placed after planning with a 3D inspection software, and the angular deviation was assessed by evaluating the angle difference. Virtual points were created on the sleeve indicator and the tip of the placed implant fixture, and the apical deviation was assessed by measuring the distance between the two points (Figure 3D).

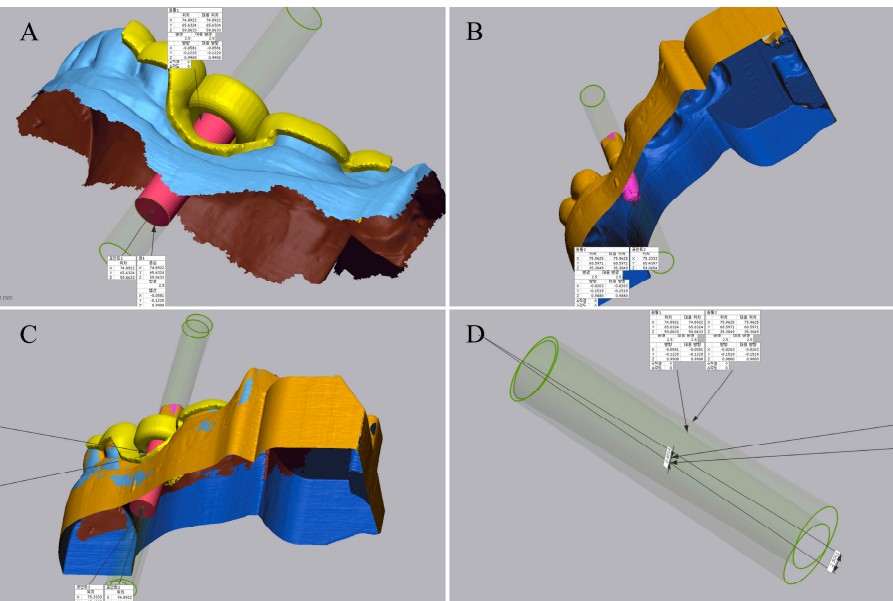

**Figure 3.** Digital evaluation method through 3D inspection software. (**A**) Planned implant position through an overlap between the data extracted from computer-guided implant software. (**B**) Placed implant position through scan body overlap. (**C**) Overlap between the data of the planned implants and the placed implants. (**D**) Deviation measurement through a virtual cylinder.

All data were analyzed using the Statistical Package for the Social Sciences (version 25.0, IBM, Chicago, IL, USA) ($\alpha = 0.05$). First, the normal distribution of data was investigated using a Shapiro–Wilk test. Equality of variance was evaluated using the Levene test for normal distribution. To compare the accuracy of the surgical guide fabricated according to software, the difference was analyzed using an independent *t*-test.

## 3. Results

For the 20 study models in which implant fixtures were placed, the depth and angle of the planned implant fixture and the placed implant fixture were compared and, as a result, the mean apical deviation of Deltanine and R2gate software was 0.603 ± 0.19 mm and 0.609 ± 0.18 mm, while the mean angular deviation of Deltanine and R2gate software was 1.97 ± 0.84° and 1.92 ± 0.52°, respectively (Tables 1 and 2). There was no significant difference between the mean apical deviation of the two software programs ($p = 0.948$), and there was no significant difference between the mean angular deviation of the two software programs ($p = 0.884$).

**Table 1.** Apical and angular deviations between the planned implant fixture and the placed implant fixture of the surgical guides fabricated with the Deltanine software.

| No. | Apical Deviation (mm) | Angular Deviation (°) |
|---|---|---|
| 1 | 0.4264 | 1.378 |
| 2 | 0.9347 | 2.8391 |
| 3 | 0.8911 | 2.3762 |
| 4 | 0.4799 | 3.213 |
| 5 | 0.6554 | 1.102 |
| 6 | 0.6141 | 3.0092 |
| 7 | 0.3532 | 1.4332 |
| 8 | 0.6747 | 2.1232 |
| 9 | 0.4748 | 1.2342 |
| 10 | 0.5325 | 0.9968 |
| Mean | 0.60368 | 1.9704 |
| Standard Deviation | 0.19182 | 0.8465 |

**Table 2.** Apical and angular deviations between the planned implant fixture and the placed implant fixture of the surgical guides fabricated with the R2gate software.

| No. | Apical Deviation (mm) | Angular Deviation (°) |
|---|---|---|
| 1 | 0.6219 | 1.9113 |
| 2 | 0.4728 | 1.3134 |
| 3 | 0.9832 | 2.983 |
| 4 | 0.7261 | 2.0142 |
| 5 | 0.6281 | 1.8602 |
| 6 | 0.3729 | 1.4312 |
| 7 | 0.5812 | 1.9786 |
| 8 | 0.4981 | 1.6823 |
| 9 | 0.3827 | 1.4821 |
| 10 | 0.8273 | 2.5829 |
| Mean | 0.6094 | 1.9239 |
| Standard Deviation | 0.1845 | 0.5207 |

## 4. Discussion

The location of the planned implants and placed implants have traditionally been evaluated by comparing the pre- and post-operative CBCT images. In addition, the image artifacts and patient movements can affect the quality of the CBCT images and limit their accuracy. This study evaluated the accuracy of computer-assisted implant surgical guides made with personal 3D printers using a

non-radiographic method that overlapped the scan image of the study model with the scan body and the scan body-implant fixture complex, respectively. In 2019, Tang et al. compared the traditional radiographic method of evaluating the implant position using CBCT, with the non-radiographic method of overlapping the scan data and inferring the implant position, and reported that there were no statistically significant differences between these methods [20]. Therefore, the evaluation of the implant position in this study is acceptable.

The positioning accuracy of the implant surgical guides can be determined by evaluating the deviation between the planned implant position and placed implant position [28]. These deviations represent different values in various papers [14]. However, a 1° deviation of the insertion angle makes an apical deviation of 0.34 mm based on a 10 mm implant fixture. In other words, a 5° deviation of the insertion angle represents an apical deviation of 1.7 mm. If the space between the implant and the adjacent tooth root is 1.5 mm, an angular deviation of 5° implies that it can invade the adjacent tooth root [29]. When considering important anatomical structures such as the inferior alveolar nerve, implant surgical guides have an allowable vertical and angle deviation of up to 1.5 mm and 3°, respectively [30]. Therefore, the vertical deviation and angle deviation in this study are within clinical tolerance.

As a result of placing implant fixtures in each of the 20 study models, there was no surgical guide fabricated with the two software programs outside the vertical deviation of 1.5 mm, and there was no statistically significant difference between the two software programs ($p = 0.948$). One surgical guide fabricated with the Deltanine software was out of tolerance of the 3° angle deviation, but there was no statistically significant difference between the two software programs ($p = 0.884$). In light of these results, it can be concluded that the surgical guide fabricated according to the software showed no difference in the positioning accuracy of the implants. The Deltanine software used in this study was compared to the R2gate software, which has been verified in the accuracy of implant placement in a previous study [17]. Although not compared to traditional implant surgical guides, the Deltanine software has been verified in comparison to the R2gate software and verified in a previous study.

The 20 resin study models used in this study had a rigid cortical bone. The inside surface was removed and filled with a mixture of orthodontic resin and sawdust, which had a lower physical property, thereby forming the cancellous bone. Since the thickness of the cortical bone varies in the clinical setting, this study also arbitrarily set the thickness of the cortical bone. In the case of a rigid personal 3D printer resin, the drill and fixture got stuck in the hard resin during the drilling process resulting in an error. In addition, fabrication of the implant surgical guides with personal 3D printers was thought to cause internal errors due to the removal of undercuts and nodules and the proficiency of the surgeon who placed the implants. Given that this was an in-vitro study measuring the accuracy of implant surgical guides made with personal 3D printers, errors above the mean can be fatal in the clinical setting. However, since the cortical bone has lower strength and higher elasticity compared to hard resin, errors due to the drill would be smaller when the implant is placed. Although the study model was fabricated in consideration of the actual clinical environment, further studies are needed to apply it to patients in actual clinical trials.

## 5. Conclusions

Based on the findings of this in-vitro study, the following conclusions were drawn:

1. The surgical guide fabricated according to the two software programs shows no difference in the positioning accuracy of the implants.
2. The accuracy of the personal 3D printed implant surgical guides is in the average range allowed by the dental clinician.
3. The surgical guide fabricated by the method presented in this study can be utilized in dental clinical practice.

**Author Contributions:** Conceptualization, S.-M.K.; methodology, K.S. and D.-Y.K.; validation, K.-B.L.; formal analysis, D.-Y.K.; investigation, K.S.; data curation, S.-M.K.; writing—original draft, S.-M.K.; visualization, K.S.; supervision, K.-B.L.; project administration, K.-B.L.

**Funding:** This research was supported by the Ministry of Trade, Industry & Energy (MOTIE, Korea) under the Industrial Technology Innovation Program (No. 10062635); and the Institute for Information & Communications Technology Promotion (IITP) for a grant funded by the Korea government (MSIP) (B0101-19-1081); and Korea Institute for Advancement of Technology (KIAT) through the National Innovation Cluster R&D program (P0006691); the Technology Innovation Program (10077743, Development of handpiece design for air turbine and root canal treatment) funded By the Ministry of Trade, Industry & Energy(MOTIE, Korea).

**Acknowledgments:** The authors thank the researchers of the Advanced Dental Device Development Institute, Kyungpook National University for their time and contributions to the study.

**Conflicts of Interest:** The authors declare no conflicts of interest. The funders had no role in the design of the study; in the collection, analyses, or interpretation of the data; in the writing of the manuscript, or in the decision to publish the results.

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
