# Peer review of "Digital Evaluation of the Accuracy of Computer-Guided Dental Implant Placement: An In Vitro Study"

_applsci, doi:10.3390/app9163373_

Round 1

Reviewer 1 Report

Dear authors:

Congratulations for the research done.

I hope to see some "in vivo" research done with this analysis.

I have no need for any change on this paper.

best regards

Author Response

We are grateful to the reviewers for their critical comments and useful suggestions that have helped us to greatly improve our paper. As indicated in the following responses, we have reflected all these comments in the revised version of our paper. Furthermore, we have had the manuscript checked by a professional English editing service.

Reviewer #2

Congratulations for the research done.

I hope to see some "in vivo" research done with this analysis.

I have no need for any change on this paper.

best regards

Response: Thank you for your opinion.

Reviewer 2 Report

Authors evaluated the accuracy of computer-assisted implant surgical guides which were designed using implant planning CAD software. However, They did not compare the traditional implant surgical guides as control. The authors should compare at least two groups. Otherwise, it could not be verified the accuracy of computer-guided implant placement.

Author Response

We are grateful to the reviewers for their critical comments and useful suggestions that have helped us to greatly improve our paper. As indicated in the following responses, we have reflected all these comments in the revised version of our paper. Furthermore, we have had the manuscript checked by a professional English editing service.

Reviewer #2

Authors evaluated the accuracy of computer-assisted implant surgical guides which were designed using implant planning CAD software. However, They did not compare the traditional implant surgical guides as control. The authors should compare at least two groups. Otherwise, it could not be verified the accuracy of computer-guided implant placement.

Response: Thank you for your suggestion for improving the quality of the manuscript. We have carefully considered your comments. We have modified all of the currently submitted manuscripts with what we were preparing for the follow-up study.  Two types of implant planning CAD software were used to compare the positioning accuracy of implants. The Deltanine software used in the manuscript submitted was compared to the R2gate software, which has been verified the accuracy of implant placement in previous study. And it was statistically analyzed and compared the two software. Although not compared to traditional implant surgical guides, Deltanine software has been verified in comparison to R2gate software verified in previous study.

Reviewer 3 Report

Authors should emphasis further that how the positioning accuracy of computer guided implants can be determined and how would it effect the implant success.

Is there any limitation of vertical and angle deviations measurements while determining the implant position?

How the positioning inaccuracies of superimposing implants were rectified while collecting the data. 

Author Response

We are grateful to the reviewers for their critical comments and useful suggestions that have helped us to greatly improve our paper. As indicated in the following responses, we have reflected all these comments in the revised version of our paper. Furthermore, we have had the manuscript checked by a professional English editing service.

Reviewer #3

Authors should emphasis further that how the positioning accuracy of computer guided implants can be determined and how would it effect the implant success.

Response: Thank you for your suggestion for improving the quality of the manuscript. We have carefully considered your comments. As the reviewer notes, we have added text and references to improve the Introduction section.

“Son's study shows that positioning accuracy of computer guided implants can vary depending on experimental conditions, and accuracy of a cone beam computed tomogram (CBCT) or intraoral scanning of the patient and accuracy of the 3D printing or milling has been reported to have a significant effect on positioning of implants [17]. If the positioning accuracy of computer guided implants is inaccurate, it can lead to an unintended position, which may be the biggest cause of implant failure.”

Is there any limitation of vertical and angle deviations measurements while determining the implant position?

Response: Thank you for your accurate comment. We have measured the vertical and angle deviations using 3D inspection software (Geomagic Control X). However, there was no limitation when analyzing software for vertical and angle deviations.

How the positioning inaccuracies of superimposing implants were rectified while collecting the data.

Response: We very much appreciate the reviewer’s comment and respect the reviewer’s insight. We have superimposed the scan data and analyzed vertical and angle deviations using 3D inspection software. The inaccuracy of the overlapping process was overcome in the following methods: (1) Precise scan data was obtained by calibration of the 3D scanner. (2) The operator performed the evaluation by confirming the exact overlapping in the 3D inspection software. As the reviewer notes, we have added text and references to improve the Materials and Methods section.

“To evaluate the apical deviation and angular deviation, 3D inspection software (Geomagic control X, 3D system, Morrisville, USA) was used, and the scan, design, and sleeve indicator STL files were extracted from the implant planning CAD software and overlapped to show the position of the planned implant fixture (Fig. 3A). To confirm the position of the actual implant fixture, the study models with implant fixtures were scanned with a connected scan body (Osstem, Busan, Korea) to extract STL files and were overlapped with the scanned file by connecting the scan body-implant fixture complex (Fig. 3B). Precise scan data was obtained by calibration of the 3D scanner. And the operator performed the evaluation by confirming the exact overlapping in the 3D inspection software. A virtual cylinder was created for each implant fixture placed after planning with a 3D inspection software, and the angular deviation was assessed by evaluating the angle difference. Virtual points were created on the sleeve indicator and the tip of the placed implant fixture, and the apical deviation was assessed by measuring the distance between the two points (Fig. 3D).”

Round 2

Reviewer 2 Report

No any comment for the revised version.